# *Rousettus aegyptiacus* Fruit Bats Do Not Support Productive Replication of Cedar Virus upon Experimental Challenge

**DOI:** 10.3390/v16091359

**Published:** 2024-08-26

**Authors:** Björn-Patrick Mohl, Sandra Diederich, Kerstin Fischer, Anne Balkema-Buschmann

**Affiliations:** Institute of Novel and Emerging Infectious Diseases, Friedrich-Loeffler-Institut, Suedufer 10, 17493 Greifswald, Germany; bjoern-patrick.mohl@fli.de (B.-P.M.); sandra.diederich@fli.de (S.D.); kerstin.fischer@fli.de (K.F.)

**Keywords:** Cedar henipavirus, *Rousettus aegyptiacus*, infection model, data loggers

## Abstract

Cedar henipavirus (CedV), which was isolated from the urine of pteropodid bats in Australia, belongs to the genus *Henipavirus* in the family of *Paramyxoviridae*. It is closely related to the Hendra virus (HeV) and Nipah virus (NiV), which have been classified at the highest biosafety level (BSL4) due to their high pathogenicity for humans. Meanwhile, CedV is apathogenic for humans and animals. As such, it is often used as a model virus for the highly pathogenic henipaviruses HeV and NiV. In this study, we challenged eight *Rousettus aegyptiacus* fruit bats of different age groups with CedV in order to assess their age-dependent susceptibility to a CedV infection. Upon intranasal inoculation, none of the animals developed clinical signs, and only trace amounts of viral RNA were detectable at 2 days post-inoculation in the upper respiratory tract and the kidney as well as in oral and anal swab samples. Continuous monitoring of the body temperature and locomotion activity of four animals, however, indicated minor alterations in the challenged animals, which would have remained unnoticed otherwise.

## 1. Introduction

Cedar henipavirus (CedV) belongs to the genus *Henipavirus* in the family *Paramyxoviridae*, whose most prominent representatives, Hendra virus (HeV) and Nipah virus (NiV), display a high zoonotic potential and pathogenicity [1]. Infections were first described in Australia (HeV) and Southeast Asia (NiV) in the 1990s and have caused outbreaks with severe respiratory and neurological disease in horses and humans in Australia, as well as in pigs, horses and humans, in Malaysia, Singapore, Bangladesh, India and the Philippines since then [2,3,4,5,6,7]. In humans, the fatality rate among clinical NiV cases can well exceed 70% [6,8]. Different *Pteropus* fruit bat species have been identified as the natural NiV and HeV reservoirs [9], which may, by shedding the virus via saliva or urine, cause virus transmission to humans or livestock via contaminated date palm sap or fruits [10,11,12].

In 2012, a new member of the *Henipavirus* genus was detected in urine samples of Australian fruit bats during HeV surveillance studies [13]. Although CedV is closely related to the highly pathogenic representatives NiV and HeV, certain genetic differences have been identified, which may be responsible for the low pathogenicity of this virus. As an example, CedV binds to the cellular receptor ephrin-B1 and ephrin-B2, while the other known bat-associated henipaviruses use ephrin-B2 and ephrin-B3 [14]. Furthermore, variations in the phosphoprotein P gene result in the loss of the open reading frame of the V protein, which is expressed in NiV and HeV by RNA editing. This may influence the CedV pathogenicity since the P and V proteins have been shown to interact with the host’s interferon system [13,15]. Although ferrets and hamsters only develop subclinical CedV infections [13,16], it has recently been shown that transgenic mice lacking type I interferon receptor (IFNAR KO mice) can indeed be infected [17,18]. The fact that this low-pathogenic virus displays a high degree of similarity to the otherwise highly pathogenic representatives HeV and NiV makes it a candidate as a model virus for various scientific questions. The rescue of a recombinant CedV clone has been performed under biosafety level (BSL) 2 conditions [19], and CedV has generally been assigned to BSL 2 in a number of countries, including Germany. However, the original isolate which was used in this study and which was isolated under BSL 4 conditions, must still be handled in a BSL 4 laboratory. 

Since fruit bats of the genus *Pteropus* subfamily *Pteropodinae* have been identified as the natural CedV reservoir, we were interested in establishing a CedV infection model in *Rousettus aegyptiacus* bats belonging to the subfamily *Rousettinae* in the *Pteropodidae* family. Bats of this species are being kept as a productive breeding colony at the Friedrich-Loeffler-Institut (FLI), and their susceptibility to different viral agents, including Marburg virus, SARS-CoV-2, influenza A H9N2, and Kasokero orthonairovirus has been published recently [20,21,22,23]. However, this species of Old World bats, which is distributed over most of the African continent as well as Southeast Asia and islands of the South Pacific, has recently been shown not to support a productive NiV infection [24]. Meanwhile, reports on the circulation of other henipaviruses in this species in different regions throughout Africa [25,26] motivated us to assess the susceptibility of this species to a CedV infection, which would open the possibility of working in a putative reservoir host under BSL2 conditions, if a recombinantly generated CedV is used [18]. The availability of such a model for henipavirus infections would open various possibilities for future research, as these bats are available in a number of research facilities, and the work would potentially not need to be done in a high containment facility. Based on the hypothesis that juvenile bats, as well as females during the reproductive phase, display elevated susceptibility to viral infection [27,28], we chose to use adult lactating females and their unweaned pups for the infection study, allowing us to also assess their age-dependent susceptibility to a CedV infection. 

## 2. Materials and Methods

All work involving replication competent CedV, which originally had been isolated in a BSL4 facility, was performed in FLI’s BSL4 laboratory and animal facility. This circumstance considerably reduced the number of animals that could be included in this challenge study.

### 2.1. CedV Acquisition and Propagation

Cedar virus (Genbank Accession No. JQ001776) was kindly supplied by the Australian Centre for Disease Preparedness (ACDP)/Commonwealth Scientific and Industrial Research Organisation (CSIRO) in Geelong/Australia. The virus was propagated in a BSL4 containment in Vero76 cells from FLI’s cell-culture collection in Veterinary Medicine in DMEM supplemented with 2% FCS. Viral nucleic acid was extracted and sent to Eurofins Genomics Europe Sequencing GmbH, Ebersberg, Germany, where sequencing was performed using proprietary methods using the Illumina NovaSeq platform. The virus titer of the stock solution was determined by plaque assay using Vero76 cells. Briefly, 90–100% confluent cells were inoculated with tenfold serial dilutions of the virus stock solution and incubated for 1 h at 37 °C with 5% CO_2_. Next, the inoculum was withdrawn, and cells were overlayed with 1% carboxymethylcellulose (Sigma-Aldrich, Darmstadt, Germany) in DMEM with 2.5% FCS and incubated at 37 °C with 5% CO_2_ for 4 days. After that, the overlay was carefully removed, and plates were fixed with 10% formalin for 1 h at room temperature (RT). Next, cells were washed with phosphate-buffered saline (PBS) before adding 0.5% crystal violet in 70% ethanol and incubating at RT for 15 min. Finally, wells were washed with distilled water before plaques were counted and plaque-forming units (pfu/mL) were calculated.

### 2.2. R. aegyptiacus Challenge Experiment

Bats from the *R. aegyptiacus* breeding colony at the FLI were assigned to the study groups of 2 mock-inoculated (juvenile, approximately six months old) and 8 infected bats (four adults 3 to 8 years old with their unweaned pups, 9 to 10 weeks old). Bats were intranasally inoculated with a dose of 8 × 10^4^ pfu per animal in a 150 µL volume. Animals were housed in groups of 2 animals (mock-inoculated) and 4 animals (infected) in cages of 75 × 130 × 65 cm. All animals had ad libitum access to water and fresh fruits. Clinical scores were monitored daily, whereas body weight could only be measured when bats were under short isoflurane inhalation anesthesia on days 2, 4, 7 and 10 post-infection (dpi) for the collection of oral- and anal swabs and nasal lavage samples. Handling and sampling were always performed, starting with the mock-inoculated group to minimize the contamination risk. At 2 dpi, two infected adult bats and their pups were euthanized by deep isoflurane anesthesia and cardiac exsanguination, and samples from the nasal conchae, trachea, lung, spleen, kidney, urinary bladder and brain were frozen for RNA and viral analysis. At 6 dpi, another infected adult bat with her pup were sacrificed and necropsied as described above, and the remaining adult bat and her pup, along with both mock-inoculated bats, were sacrificed and sampled at 14 dpi. This last time point was included to allow for the detection of a seroconversion.

### 2.3. Total RNA Extraction and Detection of Viral RNA by Real-Time RTq-PCR

Total RNA was extracted from anal and oral swab samples and nasal washes using the QiaAmp Viral RNA Kit (Qiagen Hilden, Hilden, Germany). Tissue samples were homogenized using a TissueLyser (Qiagen). Then, 250 µL tissue homogenate was mixed thoroughly with 750 µL Trizol LS (Thermo Fisher Scientific, Carlsbad, CA, USA). Next, 200 µL chloroform was added to the sample, mixed well, and then incubated for 10 min at room temperature (RT) before centrifugation at 7000× *g* for 10 min at 2–8 °C. The aqueous phase was mixed with an RNA carrier and 750 µL isopropanol before incubation for 10 min at RT, followed by centrifugation at 13,000× *g* for 10 min at 2–8 °C. The pellet was mixed with 500 µL 70% ethanol and centrifuged at 6000× *g* for 5 min at RT. The pellet was dried for 15 min at RT before resuspension in 50 µL RNAse free water.

For the detection of viral RNA, we established the following set of primers and probes targeting a region in the viral nucleoprotein (N) gene: TGAGGATCTTTGTTCCAGCG (CedV-N_For) and GTGACTCACGACATCCATC (CedV-N_Rev), as well as a probe with a 5′ 6Carboxyfluoresceine (FAM) reporter dye and a 3′ Black Hole Quencher (BHQ1) FAM-TCCAAACCTCAGATGGCGTT-BHQ1 (CedV-N_Probe). 

A total of 2.5 μL of RNA in a total reaction volume of 25 μL per reaction was analyzed using the QuantiTect Probe RT-PCR Kit (Qiagen). The real-time RT-qPCR was performed with a CFX96 Real-Time PCR Detection System (Bio-Rad Laboratories, Hercules, CA, USA). The cycling conditions used were as follows: 50 °C for 30 min, 95 °C for 15 min, followed by 42 cycles at 95 °C for 10 s, 54 °C for 30 s and 72 °C for 30 s. Fluorescence data were collected after each 54 °C step and analysis of the fluorescence data was conducted with the CFX Manager software version 4.1.2433.1293 (Bio-Rad Laboratories, Hercules, CA, USA).

### 2.4. Monitoring of Body Core Temperature and Locomotor Activity Using Data Loggers

As a pilot investigation to assess the value of monitoring physiological data, we measured the body core temperature and locomotor activity throughout the study from −3 dpi to 14 dpi for two animals per group (2 mock-inoculated juveniles, and one infected mother and pup) using intraperitoneally implanted DST micro-ACT data loggers (Star-Oddi, Gardabaer, Iceland). These were programmed to record temperature and acceleration-based activity every 10 min. The DST micro-ACT logger measured the acceleration in three axes using a defined sampling frequency for one minute; then, the on-board algorithm calculated several statistical parameters of the external acceleration (EA) such as minimum, maximum, average and variance of that variable. The EA is the acceleration above standard gravity defined with calibration and calculated as the vectoral sum of body acceleration (VeDBA) in milli-gravity. All activity recordings used 1 Hz sampling frequency for 1 min to determine the activity parameters apart from two hours a day where activity was recorded for 1 min at 10 Hz every 10 min. In addition, the raw accelerometry data was stored.

Data loggers were implanted seven days pre-infection. One hour pre-operation, animals were injected subcutaneously (s.c.) with 0.2 mg/kg Meloxicam for analgesia. Animals were anesthetized by isoflurane inhalation. The abdomen was opened in the linea alba and the data logger was placed into the abdomen without fixation to the muscular layer of the abdomen wall. During surgery, 5 mg/kg of Enrofloxacin was instilled intraperitoneally as an antibiotic. The muscular layer, as well as the outer skin, were closed separately using non-absorbable nylon (USP 4/0). The wounds were checked daily until the end of the experiment. During the necropsy, the data loggers were removed and disinfected before extracting the data using the communication box (Star-Oddi, Gardabaer, Iceland) and the associated Mercury 5.90 application software (Star-Oddi, Gardabaer, Iceland). Data points were graphed against time on the x-axis.

### 2.5. Serological Analysis

Serum samples collected during euthanasia of the animals were analyzed in an in-house ELISA for antibodies against the CedV glycoprotein G produced in *Leishmania tarentolae* and against the CedV N nucleoprotein expressed in *Spodoptera frugiperda* (Sf9) insect cells (FLI Collection of Cell Lines in Veterinary Medicine (CCLV)), infected with a recombinant baculovirus coding for the CedV N protein carrying an N-terminal histidine tag. This protein was generated following the protocol described for the Ebola virus nucleoprotein but using the coding sequence of the CedV N protein. Briefly, the coding sequence was cloned into the pAB-bee™-FH vector (AB vector, San Diego, CA, USA) and used with Profold™-ER1 baculovirus DNA (Ab Vector, San Diego, CA, USA) and the protein was then expressed in Sf9 insect cells [29]. Assays were performed in accordance with what has been described for NiV and HeV G proteins [30,31] and for the DIVA ELISA using the HeV G and N proteins [32]. Briefly, 100 ng protein per well was coated on a Nunc Maxisorp plate (ThermoFisher, Dreieich, Germany), and the plate was incubated at 4 °C overnight. On the following day, the plate was washed once with 150 µL PBS containing 0.05% Tween 20 (PBS-T) per well before the plate was blocked with 5% skim milk in PBS-T for 1 h at 37 °C. After washing with 150 µL PBS-T per well, 1:100 dilutions of the bat sera in 2.5% skim milk in PBS-T were added to two wells each and incubated at 37 °C for 1 h. The plate was washed three times, and peroxidase-conjugated Protein A/G (Thermo Scientific, Dreieich, Germany) diluted 1:30.000 in PBS-T was added and incubated for 1 h at 37 °C. The plate was washed again three times before TMB substrate (Bio-Rad, Munich, Germany) was added and incubated at RT for 10 min. Next, the same volume of 1 M H_2_SO_4_ was added to the wells, and the optical density (OD) was measured at 450 nm using a Tecan Infinite 200 Pro reader. Sera from a pig vaccinated with CedV G and a rabbit vaccinated with CedV N proteins were used as positive controls, and serum of an unvaccinated bat was used as the negative control. The percentage of the sample OD in relation to the respective positive control was calculated. A preliminary cut-off value of 15% was set for both assays. 

## 3. Results

Sequence analysis of the CedV stock used for the preparation of the inoculum revealed two silent mutations at positions 11777 (G->A) and 16316 (A->G), which both did not result in a change in the amino acid sequence. 

### 3.1. Clinical Signs and Virological Analysis

None of the *R. aegyptiacus* bats, independent of their age, did facilitate a productive CedV infection. RT-qPCR analysis using our in-house protocol for the detection of CedV genomic RNA was applied to test oral and anal swabs, nasal lavage, and tissue samples (nasal conchae, trachea, lung, spleen, kidney, bladder, and brain) and revealed marginal levels of viral RNA at Ct-values above 35 only in the four bats sacrificed at 2 dpi (Table 1), while the control sample was clearly positive with ct-values below 20. No other samples collected during the 14-day study indicated the presence of viral RNA. Although virus titration attempts are crucial for the demonstration of productive infection, no virus titration attempts were made in light of these very weak PCR signals.

The mock-inoculated control bats (juveniles) did not gain body mass readily, with one animal demonstrating a decreasing body mass of approximately 4% by 6 dpi (FH 2; Figure 1) before rebounding. In contrast, the pups continuously gained body mass during the 14 days of the study, whereas two adults (FH 3 and 5) initially lost body mass by 2 dpi, while the other two adults (FH 7 and 9) displayed an overall increase (Figure 1). This initial decrease in body mass in the adult bats correlates with the RT-qPCR-positive adult bats (FH 3 and 5; Table 1), which displayed a weight decrease of approximately 6%. Conversely, both pups (FH 4 and 6), where we detected borderline amounts of viral RNA by RT-qPCR, still displayed a weight gain between 4 and 7% by 2 dpi (Figure 1).

### 3.2. Serology

Serological analysis of the final serum samples collected during the necropsies of the animals at 2 dpi (two adults and their pups), 6 dpi (one adult and her pup) and 14 dpi (one adult and her pup plus two mock-inoculated juvenile bats), did not reveal the generation of CedV G- or CedV N-specific antibodies in any of the animals in relation to the values of their respective serum samples collected prior to the infection (Table 2).

### 3.3. Body Temperature and Locomotor Activity of Selected Individuals

Monitoring of body core temperature and locomotor activity of both mock-inoculated juvenile bats and one CedV-challenged adult female and her pup was carried out from −3 to 14 dpi (Figure 2 and Figure 3). All animals displayed a nocturnal circadian rhythm, with both body temperature (Figure 2) and locomotor activity (Figure 3) increasing during the nighttime when the lights were off (time period shaded in gray), while both decreased during the day phase when the lights were on (time period shaded in white). Daily animal handling during morning hours resulted in increased body temperatures in conjunction with low locomotor activity (Figure 2A and Figure 3A). In regard to body temperature, the observed handling spike was particularly pronounced for the pup that displayed some of the highest readings of approximately 40 °C on 5 and 10 dpi. It needs, however, to be mentioned that this animal had already reached temperatures of 39 °C or higher even before the infection (Figure 2A). Conversely, the mock-inoculated juveniles displayed some of the lowest body temperatures of approximately 36 °C between 3 and 9 dpi during the daytime, while the infected pup and adult remained slightly higher (Figure 2A). In regard to locomotor activity during the nighttime, the pup was the most active, while the adult was the least active, and both juveniles ranged between these two (Figure 3A). Interestingly, during daytime, the juveniles were the least active, and the pup was again the most active (Figure 3A).

Due to the low sample size used in this pilot setup, no statistical evaluation could be conducted. However, we could observe trends between the groups. During the night, the pup tended to have a slightly lower mean body temperature (37.24 °C, ±0.51) compared to the juveniles (37.62 °C, ±0.64) and the adult (37.55 °C, ±0.56) (Figure 2B). Interestingly, the corresponding mean locomotor activity of the pup was higher (89.52 milli-g, ±80.81) than that observed in the juveniles (67.18 milli-g, ±70.02) and the adult (47.46 milli-g, ±34.57) (Figure 3B).

During the day, all three groups had slightly reduced mean body temperatures than during the night. The pup had a mean body temperature of 37.09 °C (±0.71), the juveniles 36.67 °C (±0.68), and the adult 37 °C (±0.63) (Figure 2B). Similarly, the corresponding locomotor activity was also lower during the day for all three groups compared to the night, with the pup displaying a mean of 40.87 milli-g (±25.07), juveniles 18.69 milli-g (±14.84), and the adult 22.39 milli-g (±17.9) (Figure 3B).

## 4. Discussion

Our challenge of *R. aegyptiacus* bats did not induce a productive CedV infection in any of the challenged animals, irrespective of their age. Although others had reported the non-permissiveness of this bat species to a NiV infection [24], this result is still somewhat surprising since Rousettus bats belong to the same family of *Pteropodidae* as the Australian *Pteropus* ssp. bats, which have been identified as the natural CedV reservoir. These results suggest there is a narrow host range of henipaviruses, or the applied challenge route did not sufficiently mimic the natural infection. The latter hypothesis is supported by the fact that we were also unsuccessful in infecting *R. aegyptiacus* bats with the H9N2 influenza virus, although this virus had been isolated from the same species [23]. The very low levels of CedV-related RNA that we detected in the swab and tissue samples collected at 2 dpi in our study are most probably related to residual inoculum rather than the virus that replicated in these animals. If virus replication had occurred in these animals, one would expect to also find viral RNA at later time points after infection, which was not the case in any of the animals analyzed here. Thus, our results are in line with what has been recently described for an experimental challenge of *R. aegyptiacus* bats with the highly pathogenic NiV [24], where *R. aegyptiacus* bats were reported not to propagate NiV under experimental conditions. The authors of this study also conducted an amino acid sequence alignment of the cellular receptors ephrin B2 and ephrin B3 from different bat species and found no unique amino acid differences in the critical G protein-binding loop between the *R. aegyptiacus* sequence and those of permissive species [24].

On the other hand, there must be additional yet undetermined factors influencing the susceptibility to experimental challenge. Recently, we reported a study where we challenged *R. aegyptiacus* with an H9N2 influenza A isolate that had been isolated from the same bat species in Egypt and still could not induce a productive infection with virus shedding [23]. These factors may include the optimal infection dose and route, as well as a possibly very small window of virus shedding.

We were interested in whether very young animals (unweaned pups) or lactating females may display an elevated susceptibility to this experimental challenge, as this had been postulated from field studies where young animals, as well as females during the reproductive phase, were reported to have a higher probability of positive detections than adult bats outside the reproductive phase. However, from our results, we cannot make any conclusions on these correlations, since none of the infected animals supported the infection.

In order to assess the value of measuring physiological data in such an experiment, we monitored the body core temperature and locomotor activity of both mock-inoculated juveniles as well as one adult infected bat and her unweaned pup using intraperitoneally implanted data loggers. This measurement confirmed the nocturnal activity of these animals, as well as the circadian body temperature variability that we had reported earlier for *R. aegyptiacus* individuals from our breeding colony [33]. However, the variability of the temperature values observed during this study was distinctly lower (between 36.0 °C and 40.0 °C all animals combined) in comparison to the temperature values between 34.0 °C and 41.5 °C measured in animals in the aviary in our earlier study [33]. This can be explained by the movement restriction associated with housing in cages, in combination with a lower level of relaxation during the daytime resting phase in the unaccustomed environment. In all animals, we observed elevated body temperature during the day period at the time points at which the animals were handled (sample collection and/or maintenance of cages and food supply). Although these data need to be interpreted with great care due to the low number of measured animals, it seems like the mock-inoculated animals had lower mean temperatures than both infected animals. The locomotor activity profiles did not reveal any clear differences between infected and mock-inoculated animals. The most obvious observation was a higher level of locomotor activity in the pup, which was very prominent during the night and less pronounced during the day. This is most probably due to the young age of these animals and is not related to their infection status. In general, this data logger approach nicely visualized the high sensitivity of these measurements for the detection of even subtle changes in physiological behavior. We, therefore, suggest using such approaches when conducting challenge studies in wildlife animals such as bats that will not necessarily develop any distinct clinical symptoms upon virus infection, even if virus replication occurs. In a previous study using a bat-derived H9N2 influenza virus subtype, we focused our measurement on the temperature curves and did not observe any relevant deviation of the physiological oscillation ranging between 34 °C and 41 °C [23]. Although the observed oscillation range of 36 °C to 40.5 °C was narrower in this study, a comparison of the temperature curves before and after the inoculation revealed no significant alterations. In this case, the additional measurement of locomotor activity revealed behavioral differences between the different age groups used in this study and confirmed that the infection did not significantly influence the physiological parameters of the inoculated animals.

Besides the apparent resistance against a CedV infection, we did observe some marginal variances in the body weight curves between pups, juveniles, and adult *R. aegyptiacus* bats. As expected, the pups continued to gain weight steadily throughout the study, while both juvenile and adult bats lost weight until 2 dpi before starting to regain weight. However, we cannot completely rule out that this weight loss was a result of the changed housing 7 days before the start of the study when the animals were moved into cages from the aviary where the breeding colony is being kept. In addition, this was likely related to the regular handling of the animals during the study. After a few days, the animals became more accustomed to the handling and started to gain weight. This implies that regular handling of the animals during the pre-infection accommodation phase should be planned for future studies.

Taken together, although we were not able to detect a productive CedV infection in *R. aegyptiacus* bats, we were able to show a high level of sensitivity and accuracy in measuring the body core temperature and locomotor activity using data loggers, demonstrating their relevance in monitoring disease progression and behavioral changes in the absence of clinical signs in wild animals.

## Figures and Tables

**Figure 1 viruses-16-01359-f001:**
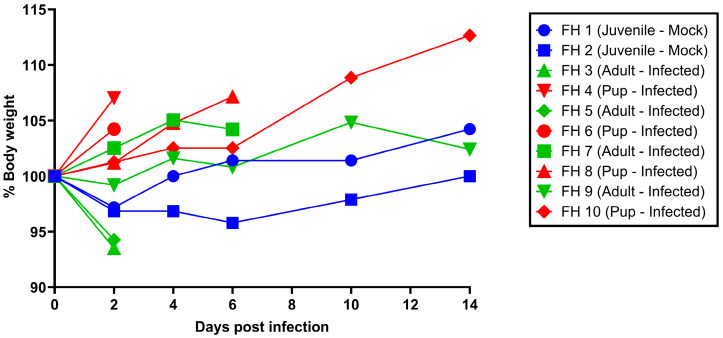
Body weight curves of *R. aegyptiacus* bats following CedV challenge. Individual weight curves of each bat: Four adult bats (green) and their four unweaned pups (red) were intranasally inoculated with 8 × 10^4^ PFU/150 µL and monitored for 14 days post-infection. Two juvenile bats (blue) were kept as a mock-inoculated control. The graph depicts the body weight changes (%) relative to 0 days post-infection.

**Figure 2 viruses-16-01359-f002:**
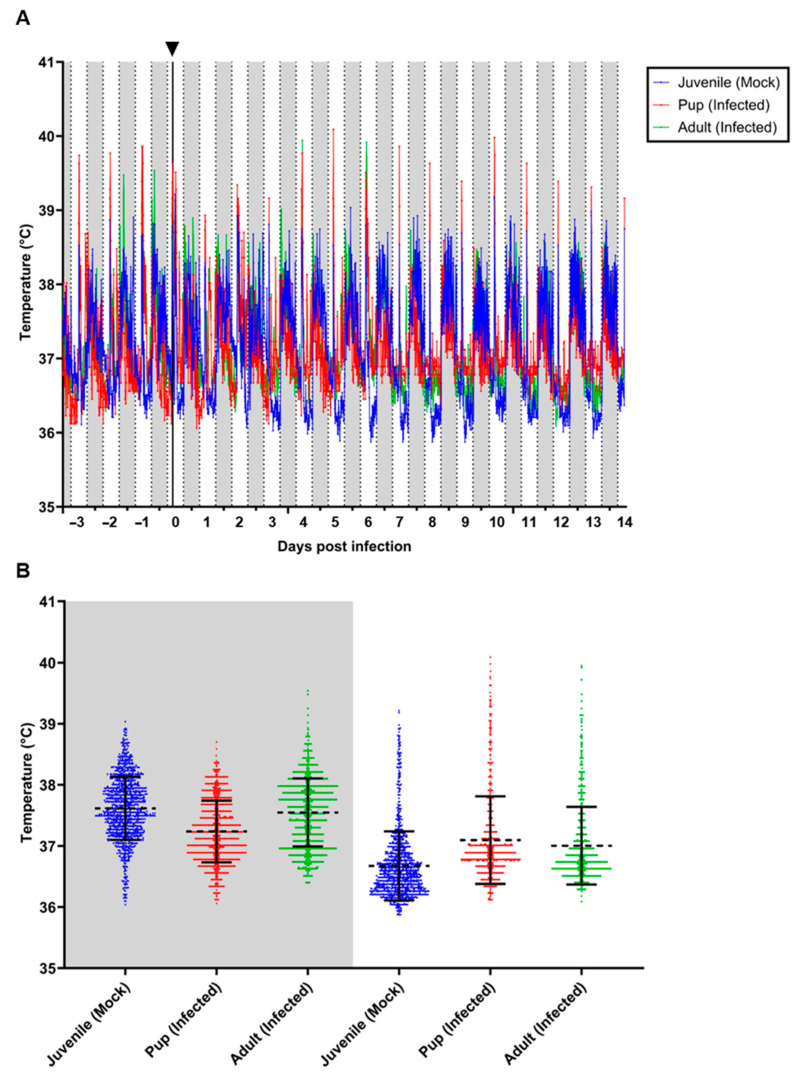
Body core temperature of two juvenile mock-inoculated *R. aegyptiacus* bats and one adult female and her pup after CedV challenge. Body core temperature was monitored at ten-minute intervals and displayed from −3 until 14 days post-infection. (**A**) Mean body core temperature of juvenile mock-inoculated bats (n = 2), adult female (n = 1) and her pup (n = 1), both CedV challenged. Daytime (6 a.m. to 6 p.m.) is marked in white, and nighttime (6 p.m. to 6 a.m.) is denoted in gray. The infection time point is shown as a black vertical line on 0 dpi, indicated by a black arrowhead on the upper horizontal axis. (**B**) Body temperature distribution during the daytime (white) and nighttime (gray), the mean is depicted by a dotted line, and error bars indicate standard deviation.

**Figure 3 viruses-16-01359-f003:**
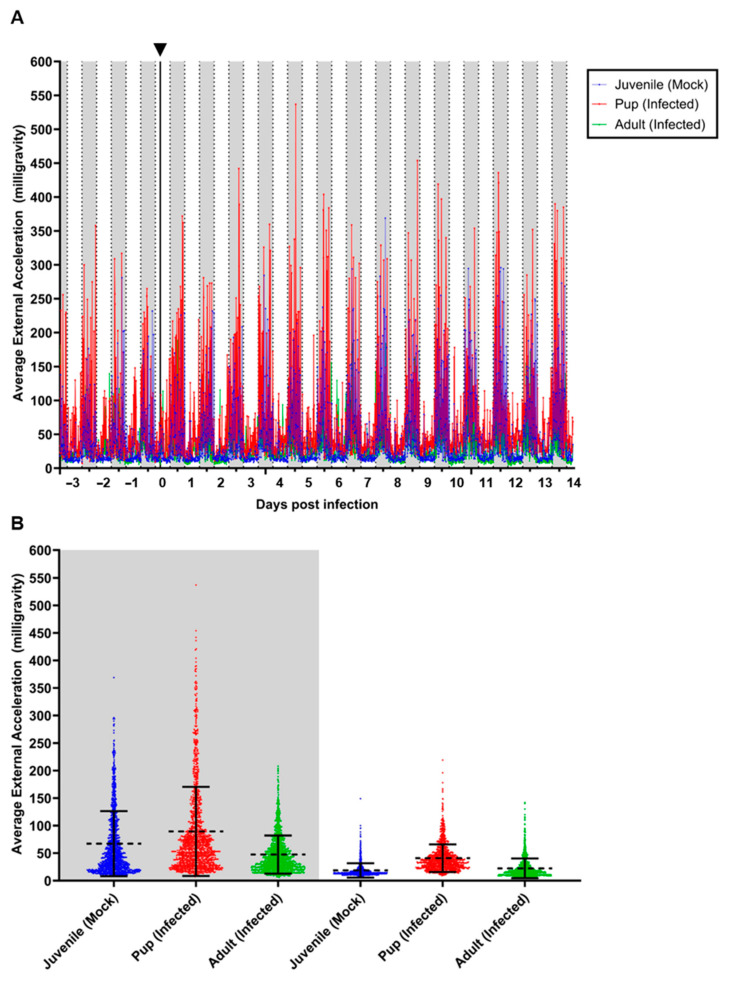
Locomotor activity of two juvenile mock-inoculated *R. aegyptiacus* bats and one adult female and her pup after CedV challenge. Locomotor activity (average external acceleration in milli-gravity) changes were monitored at ten-minute intervals and displayed from −3 until 14 days post-infection. (**A**) Mean locomotor activity. Daytime (6 a.m. to 6 p.m.) is marked in white, and nighttime (6 p.m. to 6 a.m.) is denoted in gray. The infection time point is shown as a black vertical line on 0 dpi and indicated by a black arrowhead on the upper horizontal axis. (**B**) Locomotor activity distribution during the daytime (white) and nighttime (gray), the mean is depicted by a dotted line, and error bars indicate standard deviation.

**Table 1 viruses-16-01359-t001:** Summary of positive Cedar henipavirus genomic RNA detections in all the examined samples. The Ct-value of the RT-qPCR for all positive samples is indicated.

Animal	Age Group	dpi	Sample	Ct-Value RT-qPCR
FH 3	adult	2	Nasal lavage Anal swab Organ: nasal conchae	36.97 35.11 39.2
FH 4	pup of FH 3	2	Organ: kidney	37.6
FH 5	adult	2	Organ: nasal conchae	36.9
FH 6	pup of FH 5	2	Oral swab Nasal lavage	38.72 39.13

**Table 2 viruses-16-01359-t002:** Serological analysis of serum samples using CedV G and CedV N antigens.

Animal ID and Status	PP CedV G Protein	PP CedV N Protein
FH 1 (mock-inoculated juvenile)	4.0	3.5
FH 2 (mock-inoculated juvenile)	2.9	3.4
FH 3 (inoculated adult; necropsy 2 dpi)	3.1	3.3
FH 4 (inoculated pup of FH 3; necropsy 2 dpi)	3.6	3.7
FH 5 (inoculated adult; necropsy 2 dpi)	4.6	4.1
FH 6 (inoculated pup of FH 5; necropsy 2 dpi)	4.3	6.2
FH 7 (inoculated adult; necropsy 6 dpi)	3.1	3.7
FH 8 (inoculated pup of FH 7; necropsy 6 dpi)	3.3	3.6
FH 9 (inoculated adult; necropsy 14 dpi)	4.2	3.5
FH 10 (inoculated pup of FH 9; necropsy 14 dpi)	5.1	2.9

## Data Availability

The original data presented in the study are openly available in Zenodo.

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
