# Peer review of "Rousettus aegyptiacus Fruit Bats Do Not Support Productive Replication of Cedar Virus upon Experimental Challenge"

_viruses, 2024, doi:10.3390/v16091359_

Round 1
Reviewer 1 Report
Comments and Suggestions for Authors
Mohl et al studied the susceptibility of Rousettus aegyptiacus fruit bats to Cedar virus that is closely related to highly pathogenic hendra and nipah viruses. the study is interesting but as mentioned by the authors, it would have benefited from a larger number of animals per group. This prevented them from making definite conclusion between mock and infected animals as well as between the age of subjects. Only trends were observed instead, which is of limited value. Your conclusion is also disconnected with the abstract.
Although qPCR do provide some info regarding virus shedding, and i agree that you likely retrieved a residual amount of challenge dose at day 2, an attempt to propagate virus on bat cell lines derived from kidney spleen liver and placenta would have been helpful and strengthened your findings, just like it was done to isolate the original virus from bat urine.
Specific comments:
-->abstract: there is no evidence of pathogenicity in humans. no human cases. it is unknown whether it would be. Human cell lines have been used to show susceptibility and receptor usage only. Not pathogenic in ferrets and guinea pigs. You are using CedV so you are trying to establish a fruit bat model for CedV, not HeV or NiV.
-->intro: rephrase 26-30. confusing. No NiV cases in Australia.
--> M&M: add gender (female info) in section detailing animals. tissue in formalin and Histology is mentioned but not presented in manuscript. justify timeline of 14days experiment. Add reference for primer/probe sequences. Add ref line 155. typo with Rabbit line 169.
-->Results: spell out approx. throughout the manuscript. please add cut-off value for QPCR in table 1 or M&M. INcrease symbol size fig 1. Table 2 line 212 and 216.Again power analysis would have helped on the minimum number of animals to work with. However this does not prevent the authors from comparing data between 2 groups regarding body weight and locomotor activity. likely not significant but should be done.
Author Response
Reviewer 1: Comments and Suggestions for Authors
Mohl et al studied the susceptibility of Rousettus aegyptiacus fruit bats to Cedar virus that is closely related to highly pathogenic hendra and nipah viruses. the study is interesting but as mentioned by the authors, it would have benefited from a larger number of animals per group. This prevented them from making definite conclusion between mock and infected animals as well as between the age of subjects. Only trends were observed instead, which is of limited value. Your conclusion is also disconnected with the abstract.
Author response: We thank the reviewer for their honest criticism. We are aware that the study would have benefitted from a higher animal number. However, animal experiments under BSL4 conditions often have to be reduced to the absolute minimum number of animals due to limitations in capacity concerning room availability as well as possible staff time in the facility. We have added a sentence pointing at these limitations in the Materials and Methods section (line 75 – 77).
Although qPCR do provide some info regarding virus shedding, and I agree that you likely retrieved a residual amount of challenge dose at day 2, an attempt to propagate virus on bat cell lines derived from kidney spleen liver and placenta would have been helpful and strengthened your findings, just like it was done to isolate the original virus from bat urine.
Author response: We thank the reviewer for raising this point, and we would have liked to add these analyses, as we completely agree that the detection of replication competent virus in samples from animal experiments is crucial. However, in this case with ct-values exclusively above 35, we did not expect to isolate virus from these samples and thus refrained from performing this experiment. However, we have now added a sentence explaining this in the results section (line 203 – 205): ”Although virus titration attempts are crucial for the demonstration of a productive infection, no virus titration attempts were made in the light of these very weak PCR signals.”
Specific comments:
-->abstract: there is no evidence of pathogenicity in humans. no human cases. it is unknown whether it would be. Human cell lines have been used to show susceptibility and receptor usage only. Not pathogenic in ferrets and guinea pigs. You are using CedV so you are trying to establish a fruit bat model for CedV, not HeV or NiV.
Author response: We agree that this may be misleading and we have changed the wording into: “Meanwhile, CedV is apathogenic for humans and animals. As such, it is often used as a model virus for the highly pathogenic henipaviruses HeV and NiV. In this study, we challenged eight Rousettus aegyptiacus fruit bats of different age groups with CedV in order to assess their age dependent susceptibility to a CedV infection.”
-->intro: rephrase 26-30. confusing. No NiV cases in Australia.
Author response: We have specified this as follows: “Cedar henipavirus (CedV) belongs to the genus Henipavirus in the family Paramyxoviridae, whose most prominent representatives Hendra virus (HeV) and Nipah virus (NiV) display a high zoonotic potential and pathogenicity. Infections were first described in Australia (HeV) and Southeast Asia (NiV) in the 1990s,…”.
--> M&M: add gender (female info) in section detailing animals.
Author response: We can only give this information for the adult females (already mentioned in the text), while it was too early to determine the gender for the 4 pups and 2 juveniles.
tissue in formalin and Histology is mentioned but not presented in manuscript.
Author response: Thank you for this comment, we would have overlooked this. Since we could only detect trace amounts of viral RNA in two nasal conchae samples and one kidney sample, we did not see a chance of detecting viral proteins in these tissue samples, and therefore did not attempt doing so. We therefore delete the mention of histology and formalin fixed samples from the manuscript.
justify timeline of 14days experiment.
Author response: We expected the bats to clear the virus rapidly (within 4-5 days maximum). However, we were interested to know whether a seroconversion would become detectable, and for that reason we kept the animals for 14 days. We therefore added this sentence in lines 112-113: “This last time point was included to allow for the detection of a seroconversion.”
Add reference for primer/probe sequences.
Author response: This PCR protocol hasn’t been published yet. We highlighted this by changing the word ‘used’ into ‘established’ in line 125, and we added a sentence referring to the values of the positive control to show that the protocol is capable of detecting low amounts of CedV RNA: “RT-qPCR analysis using our inhouse protocol for the detection of CedV genomic RNA was applied to test oral and anal swabs, nasal lavage, and tissue samples (nasal conchae, trachea, lung, spleen, kidney, bladder, and brain), and revealed marginal levels of viral RNA at Ct-values above 35 only in the four bats sacrificed at 2 dpi (Tab. 1), while the positive control sample was clearly positive with a ct-values below 20.”
Add ref line 155.
Author response: The expression of this particular protein has not been published yet, we appreciate the thorough assessment of our manuscript. We now refer to a publication where the expression of the Ebola virus nucleoprotein is described using the same approach. We followed this protocol, but used the sequence of the CedV N protein. We therefore added this sentence in lines 169 - 174: “This protein was generated following the protocol described for the Ebola virus nucleoprotein, but using the coding sequence of the CedV N protein. Briefly, the coding sequence was cloned into the pAB-bee™-FH vector (AB vector, San Diego, USA) and used with Profold™-ER1 baculovirus DNA (Ab Vector, San Diego, USA) and the protein was then expressed in Sf9 insect cells [29].”
typo with Rabbit line 169.
Author response: corrected
-->Results: spell out approx. throughout the manuscript.
Author response: done
please add cut-off value for QPCR in table 1 or M&M.
Author response: We considered samples with ct values higher than 38 as questionable, that is why we refer to the PCR results as ‘marginal levels of viral RNA’. Given that no ct values were detectable at all in the mock-infected animals, we decided to display these results, but with a cautious interpretation.
increase symbol size fig 1.
Author response: done
Table 2 line 212 and 216.Again power analysis would have helped on the minimum number of animals to work with. However this does not prevent the authors from comparing data between 2 groups regarding body weight and locomotor activity. likely not significant but should be done.
Author response: We considered this analysis not helpful because the animals were of different age groups (adult, pups, juveniles) and were sacrificed at different dates. Since the age does have an influence on these physiological data, we decided to only show the data trends without applying statistical tests, because that may be misinterpreted.
Reviewer 2 Report
Comments and Suggestions for Authors
In this study, Egyptian fruit bats (R. aegyptiacus) were evaluated as a potential reservoir host and experimental model for Cedar virus infection. Bats in different age groups were inoculated intranasally with Cedar virus, but no evidence of productive infection could be obtained. These results align with those obtained by others with Nipah virus infection of Egyptian fruit bats.
Although the overall result was negative, this study does contribute some important knowledge to the field. In some instances this could be communicated more effectively by the authors.
1. The overall hypothesis for the work, as articulated in the last paragraph of the introduction, needs to directly address the fact that Egyptian fruit bats were already found to to incapable of supporting productive Nipah virus replication. The hypothesis needs to explain why it is plausible that the infections of this study might lead to a different result. Is the hypothesis that Cedar virus will be different from Nipah virus in this respect? Or is the hypothesis that the use of juvenile bats will allow observation of productive infections that would otherwise be missed? It should be made more clear.
2. The potential significance of having an Egyptian fruit bat model for henipavirus infection should be emphasized more. Even though the results turned out the other way, it provides a strong justification for having tried.
3. In this study, Cedar virus was found to recapitulate the biology of Nipah virus with respect to host range among fruit bats. This could be emphasized more. If one were to make a list of observations supporting the use of Cedar virus as a valid BSL-2 model for the BSL-4 henipaviruses, there should now be one more data point to add to the list as a result of this study.
Author Response
Reviewer 2: Comments and Suggestions for Authors
In this study, Egyptian fruit bats (R. aegyptiacus) were evaluated as a potential reservoir host and experimental model for Cedar virus infection. Bats in different age groups were inoculated intranasally with Cedar virus, but no evidence of productive infection could be obtained. These results align with those obtained by others with Nipah virus infection of Egyptian fruit bats.
Although the overall result was negative, this study does contribute some important knowledge to the field. In some instances this could be communicated more effectively by the authors.
Author response: We thank the reviewer for their positive evaluation of our manuscript.
- The overall hypothesis for the work, as articulated in the last paragraph of the introduction, needs to directly address the fact that Egyptian fruit bats were already found to be incapable of supporting productive Nipah virus replication. The hypothesis needs to explain why it is plausible that the infections of this study might lead to a different result. Is the hypothesis that Cedar virus will be different from Nipah virus in this respect? Or is the hypothesis that the use of juvenile bats will allow observation of productive infections that would otherwise be missed? It should be made more clear.
Author response: We thank the reviewer for this helpful suggestion. We have revised the paragraph in the introduction accordingly. Although R. aegyptiacus bats did not support a productive NiV infection in the study published by Seifert et al., the reports on the circulation of other henipaviruses in these bats in African countries justifies in our view this challenge experiment. Our primary aim was to assess the permissiveness of this species to a CedV infection, and our second goal was to assess the age dependence of this susceptibility.
- The potential significance of having an Egyptian fruit bat model for henipavirus infection should be emphasized more. Even though the results turned out the other way, it provides a strong justification for having tried.
Author response: We added a sentence in the introduction (lines 65 – 68) stating that: “The availability of such a model for henipavirus infections would open various possibilities for future research, as these bats are available in a number of research facilities, and the work would potentially not need to be done within a high containment facility.”
- In this study, Cedar virus was found to recapitulate the biology of Nipah virus with respect to host range among fruit bats. This could be emphasized more. If one were to make a list of observations supporting the use of Cedar virus as a valid BSL-2 model for the BSL-4 henipaviruses, there should now be one more data point to add to the list as a result of this study.
Author response: We agree that our results may well be due to a narrow CedV host range. On the other hand, the challenge route may also differ too much from the natural transmission, which is supported by our earlier findings. We therefore added this paragraph in the discussion (lines 297 – 301): “These results either implicate a narrow host range of henipaviruses, or the applied challenge route did not sufficiently mimic the natural infection. The latter hypothesis is supported by the fact that we were also unsuccessful in infecting R.aegyptiacus bats with H9N2 influenza virus, although this virus had been isolated from the same species [19].”
Reviewer 3 Report
Comments and Suggestions for Authors
Dear the Authors,
Great to see this study. I hope to see that paper release soon. However, I do have one question and the other follow up the paper. Did the authors do endpoint ELISA and VNT to check the titration of the sera? The dilution 1:100 sera in the ELISA test shows the sample is positive at that dilution. Regarding the real-time PCR result, I agree with the author confirming the residual of the virus at DPI2 after inoculating the virus. Do the authors plan to repeat the experiment with intramuscular inoculation in the following study?
Thank you
Author Response
Reviewer 3: Comments and Suggestions for Authors
Dear the Authors,
Great to see this study. I hope to see that paper release soon. However, I do have one question and the other follow up the paper. Did the authors do endpoint ELISA and VNT to check the titration of the sera? The dilution 1:100 sera in the ELISA test shows the sample is positive at that dilution.
Author response: This is a valid point if positive sera are detected, as it helps to quantify the serological reaction. However, given that all sera analyzed in this study were negative in the 1:100 dilution, we did not perform such titration experiments.
Regarding the real-time PCR result, I agree with the author confirming the residual of the virus at DPI2 after inoculating the virus. Do the authors plan to repeat the experiment with intramuscular inoculation in the following study?
Author response: This could be considered for a follow-up study. As mentioned in our comment to reviewer # 2, the inoculation route which we used may not have sufficiently mirrored the natural transmission. However, we are not aware of indications that CedV (or NiV) are transmitted between bats by bites, making the intramuscular injection not very likely to be more effective than intranasal inoculation.
Reviewer 4 Report
Comments and Suggestions for Authors
Even though this is essentially negative data, it is nevertheless very important. The manuscript would benefit by deeper hypothesis explanation and contrast their study to Seifert, S. N., et al, (citation #27). Also, the results of the histology should be presented. Overall the data set and manuscript are somewhat premature.
Although the animal protocol approval and its description are included in the Institutional Review Board statement, the authors should also include some of these details in the Materials and Methods section, specifically stating also the source and biosafety containment level required/used for the wild-type Cedar virus isolate, obtained from the ACDP-CSIRO, for its propagation and analysis (BSL-4); as well as the biosafety containment level used for the animal infection studies (BSL-4). Indeed, by further detailing this biocontainment restriction it would help justify the small number of animal subjects used in this pilot study.
Ideally, the data set should be held, and a follow-up experiment with a larger inoculum and alternative route of inoculum should be performed; or better that multiple routes of inoculum including an intraperitoneal route be used. The challenge dose here is too low.
Lines 56-60; the suggestion and used of recombinant Cedar virus at BSL-2 containment is well-known to the field, as is the generation of multiple recombinant Cedar viruses and reporter gene versions that are used at BSL-2. The authors might consider to include: (Biosafety in Microbiological and Biomedical Laboratories (BMBL) 6th Edition Revised June 2020; https://www.cdc.gov/labs/pdf/SF__19_308133-A_BMBL6_00-BOOK-WEB-final-3.pdf). These latest guidelines along with including a proper citation from the literature in this regard; the BSL-2 based rescue and full characterization of recombinant Cedar virus and its ephrin entry receptor use profile are supportive to the author's argument and suggestion: (Proc Natl Acad Sci U S A. 2019 Oct 8;116(41):20707-20715. doi: 10.1073/pnas.1911773116) (Virol J. 2018 Mar 27;15(1):56. doi: 10.1186/s12985-018-0964-0). There are other published reports as well.
The abstract is unclear somewhat from the manuscript findings. The statement “Meanwhile, CedV apparently harbors a low pathogenicity for humans and animals. As such, it is often used as a model virus for the highly pathogenic henipaviruses HeV and NiV.” There is no published evidence yet of Cedar virus infection in humans, not even serological data. Also, it is not just ‘apparent’ that Cedar virus is non-pathogenic in several animal models used for Nipah or Hendra; is has been convincingly demonstrated that Cedar is non-pathogenic in those models. Also, Cedar virus has not ‘often’ been used as a model for the highly pathogenic Nipah and Hendra viruses; there is very little published data on this, and the authors overlook a recent relative model demonstrating productive Cedar virus infection and replication that requires the use of an immunocompromised mice (Front Chem Biol. 2024:3:1363498. doi: 10.3389/fchbi.2024.1363498. Epub 2024 Mar 18.).
Similarly, “Continuous monitoring of the body temperature and locomotion activity of four animals however indicated minor alterations in the infected animals which would have remained unnoticed otherwise.” The conclusion of the study is that these animals did not become infected, so really this statement should “in the Cedar virus challenged animals…..”
There is no data or summary of histology findings, although the methods state that a number of samples were obtained. This should be expanded.
Author Response
Reviewer 4: Comments and Suggestions for Authors
Even though this is essentially negative data, it is nevertheless very important. The manuscript would benefit by deeper hypothesis explanation and contrast their study to Seifert, S. N., et al, (citation #27). Also, the results of the histology should be presented. Overall the data set and manuscript are somewhat premature.
Author response: We thank the reviewer for their evaluation. We have added reference to Seifert et al in the introduction (line 59-61) and in the discussion (line 294 – 295). The reference to histology has now been deleted from the manuscript (see our response to reviewer #1).
Although the animal protocol approval and its description are included in the Institutional Review Board statement, the authors should also include some of these details in the Materials and Methods section, specifically stating also the source and biosafety containment level required/used for the wild-type Cedar virus isolate, obtained from the ACDP-CSIRO, for its propagation and analysis (BSL-4); as well as the biosafety containment level used for the animal infection studies (BSL-4). Indeed, by further detailing this biocontainment restriction it would help justify the small number of animal subjects used in this pilot study.
Author response: We are very thankful for this comment, and we added a sentence at the beginning of the Materials and Methods section: “All work including replication competent CedV, which originally has been isolated in a BSL4 facility, was performed in FLI’s BSL4 laboratory and animal facility. This circumstance considerably reduced the number of animals that could be included in this challenge study.”
Ideally, the data set should be held, and a follow-up experiment with a larger inoculum and alternative route of inoculum should be performed; or better that multiple routes of inoculum including an intraperitoneal route be used. The challenge dose here is too low.
Author response: We are considering a follow-up study using different infection routes, as we agree that the applied inoculation route may not have been sufficiently close to the yet largely unknown natural transmission route. However, in case an intraperitoneal challenge turned out to result in a productive infection, this would not mirror the natural transmission route and would therefore not necessarily show the natural pathogenesis of infection. Therefore, such a follow-up study must be carefully considered in the light of animal welfare and the 3R principle. This led us to publishing these data first, also to avoid the repetition of the same experiment in a different laboratory.
Lines 56-60; the suggestion and used of recombinant Cedar virus at BSL-2 containment is well-known to the field, as is the generation of multiple recombinant Cedar viruses and reporter gene versions that are used at BSL-2. The authors might consider to include: (Biosafety in Microbiological and Biomedical Laboratories (BMBL) 6th Edition Revised June 2020; https://www.cdc.gov/labs/pdf/SF__19_308133-A_BMBL6_00-BOOK-WEB-final-3.pdf). These latest guidelines along with including a proper citation from the literature in this regard; the BSL-2 based rescue and full characterization of recombinant Cedar virus and its ephrin entry receptor use profile are supportive to the author's argument and suggestion: (Proc Natl Acad Sci U S A. 2019 Oct 8;116(41):20707-20715. doi: 10.1073/pnas.1911773116) (Virol J. 2018 Mar 27;15(1):56. doi: 10.1186/s12985-018-0964-0). There are other published reports as well.
Author response: References to the rescue of the recombinant Cedar virus under BSL3 conditions have been added. While CedV is legally classified a BSL2 agent in Germany, this is to our knowledge not the case in all countries. We therefore still kept the wording open.
The abstract is unclear somewhat from the manuscript findings. The statement “Meanwhile, CedV apparently harbors a low pathogenicity for humans and animals. As such, it is often used as a model virus for the highly pathogenic henipaviruses HeV and NiV.” There is no published evidence yet of Cedar virus infection in humans, not even serological data. Also, it is not just ‘apparent’ that Cedar virus is non-pathogenic in several animal models used for Nipah or Hendra; is has been convincingly demonstrated that Cedar is non-pathogenic in those models.
Author response: We thank the reviewer for this helpful comment. This point has also been raised by reviewer # 1 and has been answered there.
Also, Cedar virus has not ‘often’ been used as a model for the highly pathogenic Nipah and Hendra viruses; there is very little published data on this, and the authors overlook a recent relative model demonstrating productive Cedar virus infection and replication that requires the use of an immunocompromised mice (Front Chem Biol. 2024:3:1363498. doi: 10.3389/fchbi.2024.1363498. Epub 2024 Mar 18.).
Author response: We appreciate this, and we apologize for not including the reference earlier. This has now been added to the introduction (lines 43 -46).
Similarly, “Continuous monitoring of the body temperature and locomotion activity of four animals however indicated minor alterations in the infected animals which would have remained unnoticed otherwise.” The conclusion of the study is that these animals did not become infected, so really this statement should “in the Cedar virus challenged animals…..”
Author response: This has been corrected.
There is no data or summary of histology findings, although the methods state that a number of samples were obtained. This should be expanded.
Author response: This point has been raised by other reviewers as well and it has already been answered. The reference to histology has been deleted from the manuscript.
Round 2
Reviewer 1 Report
Comments and Suggestions for Authors
thank you for addressing my comments.